# Impaired Antibody Response Is Associated with Histone-Release, Organ Dysfunction and Mortality in Critically Ill COVID-19 Patients

**DOI:** 10.3390/jcm11123419

**Published:** 2022-06-14

**Authors:** Rickard Lagedal, Oskar Eriksson, Anna Sörman, Joram B. Huckriede, Bjarne Kristensen, Stephanie Franzén, Anders Larsson, Anders Bergqvist, Kjell Alving, Anders Forslund, Barbro Persson, Kristina N. Ekdahl, Pablo Garcia de Frutos, Bo Nilsson, Gerry A. F. Nicolaes, Miklos Lipcsey, Michael Hultström, Robert Frithiof

**Affiliations:** 1Department of Surgical Sciences, Anaesthesia and Intensive Care, Uppsala University, 752 36 Uppsala, Sweden; stephanie.franzen@surgsci.uu.se (S.F.); miklos.lipcsey@surgsci.uu.se (M.L.); michael.hultstrom@mcb.uu.se (M.H.); robert.frithiof@surgsci.uu.se (R.F.); 2Department of Immunology, Genetics and Pathology, Uppsala University, 752 36 Uppsala, Sweden; oskar.eriksson@igp.uu.se (O.E.); anna.sorman@igp.uu.se (A.S.); barbro.persson@igp.uu.se (B.P.); kristina.nilsson_ekdahl@igp.uu.se (K.N.E.); bo.nilsson@igp.uu.se (B.N.); 3Department of Medical Biochemistry and Microbiology, Uppsala University, 752 36 Uppsala, Sweden; 4Department of Biochemistry, Cardiovascular Research Institute Maastricht (CARIM), Maastricht University, 6211 LK Maastricht, The Netherlands; j.huckriede@maastrichtuniversity.nl (J.B.H.); g.nicolaes@maastrichtuniversity.nl (G.A.F.N.); 5Thermo Fisher Scientific, 3450 Allerod, Denmark; bjarne.kristensen@thermofisher.com; 6Department of Medical Sciences, Uppsala University, 752 36 Uppsala, Sweden; anders.larsson@akademiska.se; 7Department of Medical Sciences, Section of Clinical Microbiology, Uppsala University, 752 36 Uppsala, Sweden; anders.bergqvist@akademiska.se; 8Clinical Microbiology and Hospital Infection Control, Uppsala University Hospital, 752 36 Uppsala, Sweden; 9Department of Women’s and Children’s Health, Uppsala University, 752 36 Uppsala, Sweden; kjell.alving@kbh.uu.se (K.A.); anders.forslund@kbh.uu.se (A.F.); 10Linneus Centre for Biomaterials Chemistry, Linneus University, 392 31 Kalmar, Sweden; 11Department of Cell Death and Proliferation, IIBB-CSIC, IDIBAPS and CIBERCV, 08036 Barcelona, Spain; pablo.garcia@iibb.csic.es; 12Hedenstierna Laboratory, Anesthesiology and Intensive Care, Department of Surgical Sciences, Uppsala University, 752 36 Uppsala, Sweden; 13Unit for Integrative Physiology, Department of Medical Cell Biology, Uppsala University, 752 36 Uppsala, Sweden

**Keywords:** COVID-19, SARS-CoV-2, critical care, antibody response, NET, histones

## Abstract

Purpose: the pathophysiologic mechanisms explaining differences in clinical outcomes following COVID-19 are not completely described. This study aims to investigate antibody responses in critically ill patients with COVID-19 in relation to inflammation, organ failure and 30-day survival. Methods: All patients with PCR-verified COVID-19 and gave consent, and who were admitted to a tertiary Intensive care unit (ICU) in Sweden during March–September 2020 were included. Demography, repeated blood samples and measures of organ function were collected. Analyses of anti-SARS-CoV-2 antibodies (IgM, IgA and IgG) in plasma were performed and correlated to patient outcome and biomarkers of inflammation and organ failure. Results: A total of 115 patients (median age 62 years, 77% male) were included prospectively. All patients developed severe respiratory dysfunction, and 59% were treated with invasive ventilation. Thirty-day mortality was 22.6% for all included patients. Patients negative for any anti-SARS-CoV-2 antibody in plasma during ICU admission had higher 30-day mortality compared to patients positive for antibodies. Patients positive for IgM had more ICU-, ventilator-, renal replacement therapy- and vasoactive medication-free days. IgA antibody concentrations correlated negatively with both SAPS3 and maximal SOFA-score and IgM-levels correlated negatively with SAPS3. Patients with antibody levels below the detection limit had higher plasma levels of extracellular histones on day 1 and elevated levels of kidney and cardiac biomarkers, but showed no signs of increased inflammation, complement activation or cytokine release. After adjusting for age, positive IgM and IgG antibodies were still associated with increased 30-day survival, with odds ratio (OR) 7.1 (1.5–34.4) and 4.2 (1.1–15.7), respectively. Conclusion: In patients with severe COVID-19 requiring intensive care, a poor antibody response is associated with organ failure, systemic histone release and increased 30-day mortality.

## 1. Background

The ongoing COVID-19 pandemic, caused by the novel SARS-CoV-2 virus, has caused millions of deaths worldwide and left the healthcare system in many countries in the worst crisis for decades. Since the virus phenotype is novel for humans, no patients have previous antibodies specific for the virus, creating a situation where, in theory, all humans are susceptible for infection and severe disease. Despite this, the clinical course of SARS-CoV-2 infection varies substantially, from asymptomatic carriers to severe multiple organ dysfunction syndrome (MODS) and death, probably explained by individual variations in the immune response.

Several risk factors, both for the development of severe disease but also for death, have been identified [1]. Age is the strongest risk factor, but several others such as male sex, cardiovascular disease, obesity, chronic obstructive pulmonary disease (COPD), Alzheimer’s disease and genetic predisposition are now known to increase the risk of poor outcomes following COVID-19 disease [2,3,4,5,6]. Even if part of the increased risk is due to physiologic fragility, for example, very old patients have lower cardiopulmonary reserve to cope with a pneumonia regardless of the causative agent, the immune response to the infection is likely of great importance [7,8]. Several studies have described the immune response during COVID-19 and defined differences between patients developing severe disease and patients with asymptomatic or mild disease [9,10]. However, somewhat conflicting results concerning antibody responses have been presented, perhaps reflecting sampling site, varying cohorts or the timing of blood sampling [11,12,13,14,15,16]. It appears that an adequate, early response from the innate immune system, including expression of type I interferons (IFN), is important for reducing viral replication, allowing the slower adaptive immune system to become fully activated [17]. SARS-CoV-2 has the ability to suppress the expression of type I IFN, and hence inhibit the innate immune response to the virus [18]. A delayed innate immune response might also lead to a longer activation time for the adaptive immune response, since the two systems are dependent on each other for optimal function. We hypothesised that a delayed or absent adaptive immune response in critically ill patients with COVID-19 would cause higher mortality and organ failure. Several groups have also reported that SARS-CoV-2 can induce neutrophil extracellular trap (NET) formation in neutrophiles and that NET-formation could be part of the immunopathology in severe COVID-19 [19,20,21,22,23,24,25].

Our group previously reported that a weak anti-SARS-CoV-2 antibody response was associated with increased mortality in a small cohort of intensive care patients with COVID-19 disease [26]. The aim of this study is to describe the effects of an impaired antibody response, with regard to 30-day survival, organ failure and activation of other parts of the immune system, identifying key pathophysiological mechanisms in a large group of intensive care patients. 

## 2. Materials and Methods

### 2.1. Study Design

This single centre, prospective observational investigation is a sub-study of the PronMed-study, approved by the Swedish National Ethical Review Agency (EPM; No. 2020-01623). Informed consent was obtained either by the patient or by a next-of-kin if the patient was unable to receive information due to their clinical status. The Declaration of Helsinki and its subsequent revisions were followed. The protocol of the study was registered a priori (ClinicalTrials ID: NCT04316884). STROBE guidelines were followed for reporting.

### 2.2. Data Collection

All patients admitted to the central intensive care unit at Uppsala University Hospital during the first wave of the pandemic in 2020, with suspected COVID-19 infection, were screened for inclusion.

Background characteristics of the patients were obtained through patients’ electronic medical records. Clinical data were collected prospectively daily. Blood samples were taken at ICU admission and three times per week during the time patients were treated in the ICU. Simplified Acute Physiology Score 3 (SAPS3) on admission and daily Sequential Organ Failure Assessment (SOFA) score were calculated prospectively [27,28]. Acute kidney injury (AKI) was diagnosed according to the Kidney Disease: Improving Global Outcome (KDIGO) creatinine criteria [29].

### 2.3. Plasma Analyses

Peripheral blood from patients with COVID-19 was collected into EDTA- and citrate-containing tubes and plasma was separated using centrifugation at 3000× *g* for 10 min. After separation, all plasma samples were stored at −80 °C.

Complete blood cell counts (CBC), plasma C-reactive protein (CRP), procalcitonin, IL-6, fibrin D-dimer, troponin I and N-terminal pro-brain natriuretic peptide (NT-pro-BNP), kidney function tests (plasma creatinine and cystatin C), liver function tests (plasma bilirubin, alanine aminotransferase (ALT), aspartate aminotransferase (AST), alkaline phosphatase (ALP)) were performed in the hospital central laboratory. CBC was analysed on a Sysmex XN instrument (Sysmex, Kobe, Japan) while plasma CRP, ferritin, troponin I, kidney and liver markers were analysed on an Architect ci16200 (Abbott Laboratories, Abbott Park, IL, USA). IL-6 was measured by a commercial sandwich ELISA kit (D6050, R&D Systems, Minneapolis, MN, USA). IgA, IgG and IgM antibodies against SARS-CoV-2 Spike-1 protein were quantified by FluoroEnzymeImmunoassay (FEIA), Phadia AB, Uppsala, Sweden. The analyses were performed on the last sample obtained during the stay at the ICU but within 30 days from symptoms onset to maximise the probability to discover plasma-antibodies. The lower limit of detection was 5 and 20 ug/L for IgA and IgM, respectively, and 10 U/L for IgG.

Cytokine and complement analyses are described in detail in the Appendix A.

SARS-CoV-2 RNA in plasma was determined by reverse transcription qPCR as previously described [30]. For qualitative and quantitative detection of viral RNA, we used the 2019-nCoV N1 reagent set from the published protocol from the Center for Disease Control (CDC) of the United States [31]. For quantitative analysis, the ISO 13485 certified molecular standard Quantitative Synthetic SARS-CoV-2 RNA: ORF, E, N (VR-3276SD, American Tissue Type Collection) was used as external calibrator. The reaction showed linearity over 6 orders of magnitude with 10^9^ copies/mL and 300 copies/mL as the upper and lower limits of quantitative detection, respectively. The viral RNA analyses were performed at samples taken between day 1 and day 7 in the ICU.

### 2.4. Histone Analyses

The presence of histones was determined via a semi-quantitative Western blotting method as previously described [32,33]. In short, plasma was diluted 10 times and separated via SDS-PAGE gel electrophoresis (4–15% gradient gel), and transferred to a PVDF membrane (Bio-Rad Laboratories, Hemel Hempstead, UK) using semi-dry blotting. After blocking, the membranes were incubated overnight with a primary rabbit anti-histone H3 antibody (1:10,000, sc-8654-R, Santa Cruz Biotechnology, Heidelberg, Germany), followed by 1 h incubation with a secondary biotin-conjugated donkey anti-rabbit IgG antibody (1:10,000, ab97083, Abcam, Cambridge, UK), and 30 min with a streptavidin-biotin complex (1:500, Vectastain, Vector Laboratories, Burlingame, CA, USA). Histone H3 bands were visualised by the WesternBright ECL substrate (Advansta, San Jose, CA, USA) on the iBright CL1500 Imaging System (ThermoFisher Scientific, Waltham, MA, USA). The band densities were quantified by iBright Analysis Software, compared to known standard concentrations of purified calf thymus H3 (Roche, Basel, Switzerland).

### 2.5. Statistics

Categorical variables are presented as number of observations (percentage of total number of observations) and continuous variables as medians and interquartile range (IQR). Comparison between dichotomous variables were made with Pearson’s Chi2-test or Fischer’s exact test as appropriate. Continuous variables were compared with the Mann–Whitney U test. Correlation between antibody levels and SAPS3/SOFA were assessed with Spearman correlation. Analyses of survival in relation to whether patients were positive or negative for antibodies were further assessed with multiple logistic regression while controlling for age. For calculations and figures, SPSS Statistics software, version 23 (IBM) was used. *p* < 0.05 was considered significant.

## 3. Results

Between 13 March and 28 September 2020, 125 patients were included. After the exclusion of patients without verified COVID-19 infection, patients where no blood samples were obtained and patients where the diagnoses of COVID-19 were considered a secondary finding, 115 patients were included in the final analyses. The vast majority (88%) of the patients in the study were admitted during March–May.

### 3.1. Patient Characteristics

The median age for all patients was 62 years and 77% were male (Table 1). Median time from onset of symptoms until ICU admission was 10 days. Ninety percent of the total cohort developed anti-SARS-CoV-2 antibodies during their time in the ICU. Eighty-nine patients (77%) were alive after 30 days. Five (4%) of the included patients had a known immune deficiency prior to admission, either due to immune suppressive treatment or disease (previous organ transplant, lymphoma or B-cell suppressive treatment). Two of these patients did not develop antibodies and two only expressed IgM and IgM + IgG, respectively. Four of these patients were alive at 30 days from ICU admission. Thus, 110 out of 115 patients had no known reason for impaired antibody responses. For the groups with negative vs. positive SARS-CoV-2 antibodies, there was no difference in median time from ICU admission to blood sampling for antibody analyses.

### 3.2. Survival and Organ Dysfunction

Fifty-nine percent of the included patients were treated with mechanical ventilation and 15% received renal replacement therapy. Patients positive for anti-SARS-CoV2 antibodies had higher 30-day survival (which was the main outcome in the present analysis) compared to patients negative for antibodies (30-day survival for IgM 83% vs. 33%, IgG 82% vs. 53% and IgA 83% vs. 47%). As a complementary analyses, 90-day survival was analysed. This confirmed the findings with higher survival rates in the patient groups positive for anti-SARS-CoV2 antibodies. (Figure 1 and Table 2).

Patients positive for IgM also had more ICU-free days, ventilator-free days, renal replacement-free days and vasoactive medication-free days. For IgG and IgA, antibody-positive patients had more renal replacement-free days. (Table 3)

In a simple logistic regression model, odds ratios (confidence interval, CI) for 30-day survival were 9.4 (2.6–34.8), 4.0 (1.3–11.6) and 5.6 (1.9–15.9) for patients positive for IgM, IgG and IgA, respectively. When adjusting for age in a multiple logistic regression model, positive tests for IgM and IgG antibodies were still correlated with higher OR for 30-day survival, whereas no difference between the groups were seen for IgA. (Figure 2)

In the correlation analyses, IgA antibody concentrations correlated negatively with both SAPS3 (r = −0.233, *p* = 0.013) and maximal SOFA score (r = −0.231, *p* = 0.014). No correlation was seen between IgG and SAPS3 or SOFA whereas the levels of IgM antibodies correlated negatively with SAPS3 (r = −0.231, *p* = 0.014).

### 3.3. Plasma Biomarkers in Relation to Antibody Response

Next, we analysed plasma biomarkers and their relation to antibody response. Clinical chemistry tests and blood cell counts analysed during ICU care were extracted from the patients’ medical records, and ICU entry and peak values were calculated.

To corroborate the association between antibody response and organ support, biomarkers of organ failure were compared among antibody-negative and -positive patients. Significant associations were observed for kidney (Creatinine, Cystatin C) and cardiac biomarkers (NT-proBNP). A prominent activation of the coagulation system with elevated D-dimer and platelet counts is an important feature of severe COVID-19 [34]. However, antibody-negative subjects had significantly lower platelet counts, and a non-significant trend towards lower D-dimer levels was observed. Furthermore, antibody-negative patients had lower CRP levels, indicating an attenuation of systemic inflammation in patients with an impaired antibody response.

To further characterise the immune response in relation to antibody positivity, biomarkers of the innate immune system, including cytokines, systemic histone release and complement activation, and white blood cell differential counts were analysed.

Antibody-negative patients had significantly higher plasma levels of extracellular histones on day 1 compared to antibody-positive patients, and elevated extracellular histone levels were observed irrespective of the antibody class (Figure 3). We could not find strong associations between antibody levels in plasma and any of the analysed cytokines (Appendix A).

Complement factors and activation markers were analysed in plasma samples taken upon ICU admission (Appendix A). There was no difference in the degree of complement activation between antibody-positive and -negative patients, as measured by C3a, C3d and sC5b9 levels in plasma. In contrast, IgG and IgA antibody-negative patients had significantly lower circulating levels of intact complement factor C3 and factor B.

## 4. Discussion

The main finding of this study is that a poor antibody response is associated with increased mortality in patients with severe SARS-CoV-2-infection. Furthermore, patients with poor antibody response have higher rates of organ failure based on SAPS3 and SOFA score, even if the correlations are weak and should be interpreted with caution, and also based on the duration of organ support such as renal replacement therapy and vasoactive medication. 

These clinical associations were corroborated by biomarker analyses, where the strongest associations were seen for kidney function and cardiac biomarkers. In particular, antibody-negative patients had strongly elevated NT-proBNP levels during ICU care, which could indicate an increased rate of circulatory failure in agreement with a trend towards longer duration of vasoactive support in these patients.

The strongest signal in our cohort for the associations with outcome is seen for IgM antibodies and it is reasonable to believe that this is due to the natural course of the B-cell development in the germinal centre (GC) reaction. Additionally, IgM, in its pentameric form, has the highest complement-activating capacity among all immunoglobulin subclasses, thus it has a very high neutralising capacity [35].

There was no significant difference in the concentration of viral copies in blood between the analysed groups, although patients negative for antibodies in this study did have numerically higher levels of viral copies in blood. The non-significance might reflect a lack of power in the study, and it would be interesting to analyse this in a larger group of ICU patients. However, we have previously described a lack of strong association between viremia and organ failure in COVID-19 [30,36].

As the production of antibodies is a result of a close and simultaneous collaboration between the innate and adaptive immune systems, alterations in the innate immune system due to age, gender and/or genetic variations will skew the adaptive responses in different directions [37]. It has been previously described that COVID-19 patients who develop a mild disease responds with a fast antibody response of short duration. Patients with severe disease instead have a slower antibody response with a longer duration [9]. Measurable antigen-specific antibodies in plasma from patients are a result of a successful GC-reaction. When it comes to viral infections, the GC-reaction is dependent on antigen-specific T-cells [38]. T-cell studies on patients with mild versus severe COVID-19 have in both cases shown a robust specific T-cell response (both CD4+ and CD8+ T-cells) but the T-cell phenotypes (e.g., cytokine production and dominant T-cell subset) differ between the two disease severity groups [39,40]. It is thus plausible that differences in T-cell responses would mirror any difference in priming of the GC-reaction and hence the antibody response. From the results in our study, it seems that among patients developing severe the disease who require intensive care, patients with a higher antibody response have a better chance of survival, suggesting that an adequate response from both T- and B-cells is of great importance in the defence against SARS-CoV-2 infections. The individual differences in the total immune response to the virus causing COVID-19 might be due to several, not mutually exclusive, mechanisms, e.g., it is described that the SARS-CoV-2 virus can inhibit the initial innate immune response through suppression of type I IFN [41]. This can lead to a slower activation of the adaptive immune system. In patients with an inherent poor type I IFN-response this effect can give rise to a greater initial viral replication causing more severe disease [42]. Thus, the slow activation of the adaptive immune system may reflect an initial suboptimal innate immune response.

This hypothesis is supported by our biomarker analysis, which by several measures indicated a lower degree of inflammation in antibody-negative subjects, possibly due to an unknown inborn or acquired immune defect. 

Our complement analyses revealed no differences in complement activation between antibody-negative and -positive subjects, but they demonstrated significantly lower circulating levels of intact complement protein C3 and Factor B in antibody-negative patients. These factors are well-known to display an acute phase response pattern and are elevated during systemic inflammation, hence lower plasma levels are agreement with an attenuated inflammatory response in antibody-negative subjects. Furthermore, a recent study on bronkoalveolar lavage in patients with severe COVID-19 describes a more aggravated local immune response in the lung, suggesting plasma analyses might be misleading in compement analyses [43].

Another hypothesis is that patients with a suboptimal adaptive immune response, instead, develop a more powerful non-specific innate immune response, partly supported by previous research [44], with increased neutrophil activation and a powerful activation of the complement system [45]. We could not find any sign of increased complement activation or cytokine release in patients with poor antibody response, but those patients had more circulating cell-free histones. SARS-CoV-2 is able to induce neutrophil extracellular trap (NET) formation in healthy neutrophils, suggesting a role for NETs in driving cytokine release, respiratory failure, and microvascular injury in COVID-19 [19,20]. In this study, patients that did not express anti-SARS-CoV-2 antibodies in the ICU had increased extracellular histones in their blood at ICU admission, which may signal systemic neutrophil activation and NET formation [21]. Although we only measured histone H3, this could be seen as a proxy for all other histones. However, it is possible that the increased levels of histones are a marker of unspecific cell damage and are not linked to NET formation. Previous research shows that the number of neutrophils and markers of NETosis are elevated in severe COVID-19 patients [22], and a link has been suggested between NETosis and poor outcome [23], indicating a crucial contribution of NETs to the severity of the COVID-19 disease [24,25]. An association has also been described between anti-SARS-CoV-2 IgA2, NET formation and poor outcome [46]. As lymphopenia and inadequate T-cell responses are found to be predictors of COVID-19 severity, [47] our findings of an association between poor antibody response and amount of histones in blood in ICU patients suggests a link between poor adaptive immune activation and NET formation. Furthermore, patients with impaired antibody responses and increased number of free histones also have lower thrombocyte values. This could be an indirect sign of platelet consumption in NET-containing microthrombi as neutrophil-platelet infiltration has been observed in pulmonary autopsies from COVID-19 patients [48].

Our findings that a poor antibody response is associated with a poor outcome in ICU patients with COVID-19 could have clinical implications. If we could identify patients that are likely to have a slow antibody response early in the disease process, these might be the patients benefiting the most from immune-modulating treatment, as studies, in which patients treated as early as 72 h after COVID-19 diagnosis with convalescent plasma or monoclonal anti-SARS-CoV-2 antibodies, have shown [49,50]. In addition, specific organ failure, such as AKI, is common in severe COVID-19 and may cause long-term effects in surviving patients [51,52]. Together with common clinical variables, these data may prove useful in improving recognition of patients at risk of extra pulmonary organ failure.

The strength of this study is the large number of intensive care patients included and the high quality of clinical data prospectively collected in combination with analyses of both antibodies, cytokines, and components of the complement system. The major weakness is that samples for antibody analyses were not taken on the same day as for the other analyses; instead, we chose to analyse the last available blood sample taken in the ICU. The reason for this was to maximise the chance to have samples taken after the patients had developed an antibody response and hopefully also switched antibody class. This might have impacted the results, but there was no difference between the compared groups in median time from admission to sample day.

## 5. Conclusions

This study confirms that a weak anti-SARS-CoV-2 antibody response in intensive care patients with COVID-19 is associated with increased 30-day mortality and our results suggest that this may be due to multiple organ dysfunction and NETosis.

## Figures and Tables

**Figure 1 jcm-11-03419-f001:**
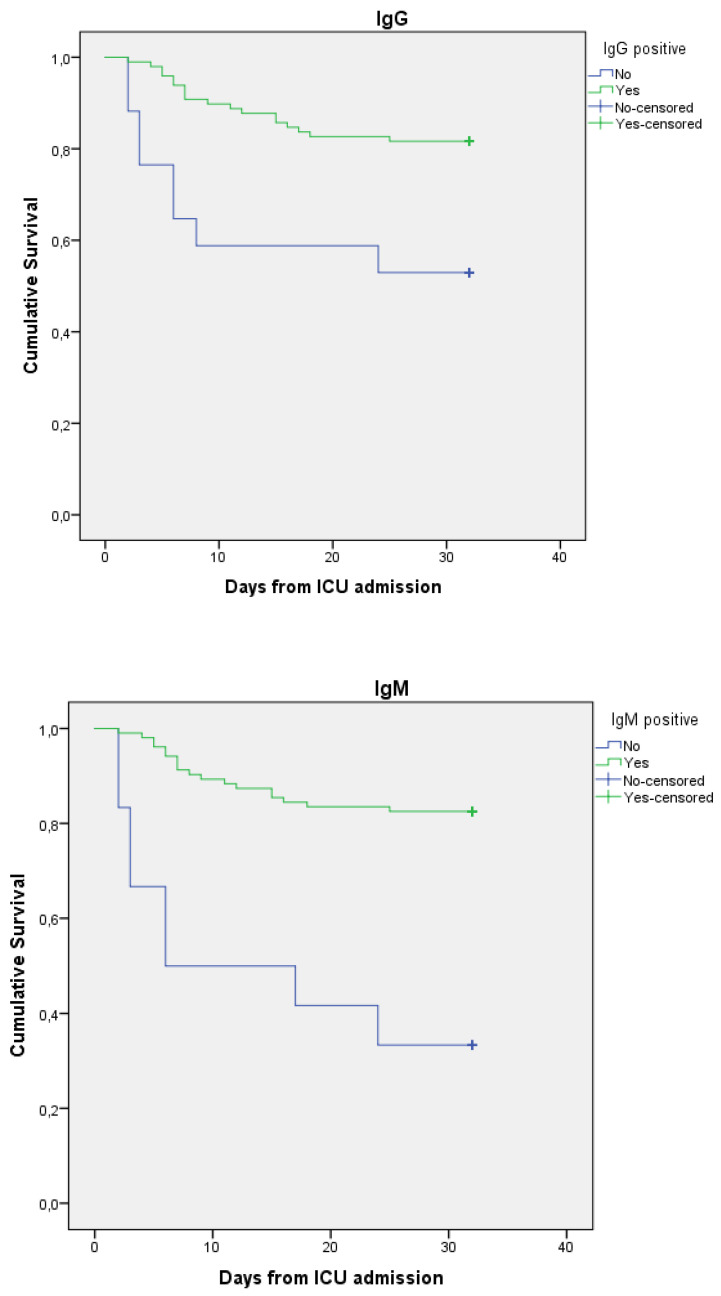
Kaplan–Meier curves describing survival after hospital admission for patients positive vs. negative for SARS-CoV-2 antibodies in plasma.

**Figure 2 jcm-11-03419-f002:**
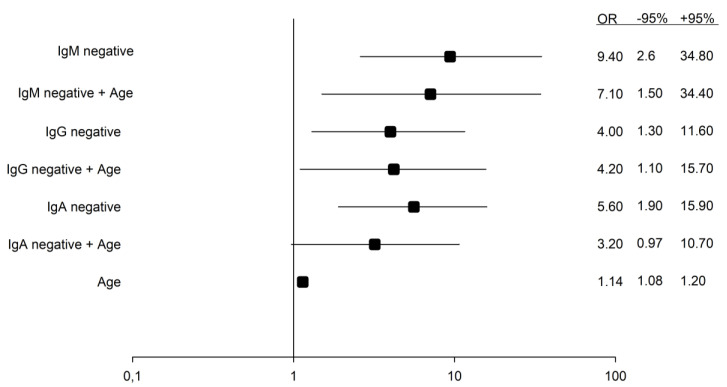
Odds ratios for death in single and multiple logistic regression models with antibody negativity and patient age as independent variables. Increasing age (counted in years) is associated with higher mortality rates.

**Figure 3 jcm-11-03419-f003:**
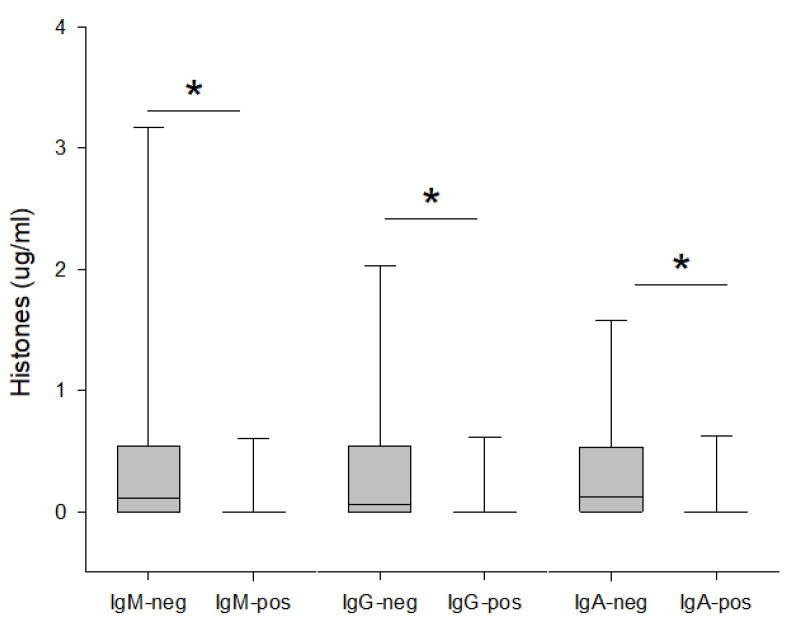
Concentration of Histones H3 in plasma in relation to positivity/negativity for SARS-CoV-2 antibodies in plasma. Antibody-positive: Before ICU discharge. Statistically significant differences between groups are marked by *.

**Table 1 jcm-11-03419-t001:** Patient characteristics.

		All Patients *n* = 115	Alive at 30 Days	
			Yes *n* = 89	No *n* = 26
Age		62 (52–71)	57 (51–67)	73 (68–79)
Male sex		88 (77%)	67 (75%)	21 (81%)
SAPS3 on ICU arrival		53 (47–57)	50.5 (46–56)	60 (55–65)
Days with symptoms on ICU arrival		10 (8–12)	10 (9–12)	10 (8–12)
BMI		28.6 (25.6–33.2)	28.8 (26.6–33.8)	27.4 (23.9–30.8)
Pulmonary disease		29 (25%)	21 (24%)	8 (31%)
Hypertension		62 (54%)	41 (46%)	21 (81%)
Diabetes		32 (28%)	24 (46%)	8 (31%)
Smoker	Ongoing	7 (6%)	5 (6%)	2 (9%)
	Previous	20 (18%)	15 (17%)	5 (23%)
Alive at 30 days		89 (77%)	89 (100%)	0 (0%)
IgG positive		98 (85%)	80 (90%	18 (69%)
IgA positive		96 (83%)	80 (90%)	16 (62%)
IgM positive		103 (90%)	85 (96%)	18 (69%)

Results are expressed as *n* (%) or median (interquartile range, IQR). Abbreviations: BMI: Body mass index (kg/m^2^), Alive at 30 days: 30 days from ICU admission. Age counted in years. Antibody-positive: Before ICU discharge.

**Table 2 jcm-11-03419-t002:** Comparison between patients with or without anti-SARS-CoV-2 antibodies.

	All Patients	Iga Positive		*p*	Igm Positive		*p*	Igg Positive		*p*
		Yes *n* = 96	No *n* = 19		Yes *n* = 103	No *n* = 12		Yes *n* = 98	No *n* = 17	
Alive at 30 days	89 (77%)	80 (83%)	9 (47%)	0.002	85 (83%)	4 (33%)	0.001	80 (82%)	9 (53%)	0.02
Alive at 90 days	84 (73%)	78 (79%)	6(38%)	0.001	82 (79%)	2 (18%)	<0.001	79 (79%)	5 (33%)	0.01
Male sex	88 (77%)	79 (82%)	9 (47%)	0.002	83 (81%)	5 (42%)	0.007	79 (81%)	9 (53%)	0.03
Thrombotic events	15 (13%)	15 (16%)	0 (0%)	n.s.	15 (15%)	0 (0%)	n.s	15 (15%)	0 (0%)	n.s.
Critical illness	15 (13%)	14 (15%)	1 (5%)	n.s.	13 (13%)	2 (17%)	n.s	14 (14%)	1 (6%)	n.s.
Secondary infection	58 (51%)	51 (54%)	7 (37%)	n.s.	52 (51%)	6 (50%)	n.s	50 (52%)	8 (47%)	n.s.
Vasoactive medication	75 (65%)	65 (68%)	10 (53%)	n.s.	67 (65%)	8 (67%)	n.s	65 (66%)	10 (59%)	n.s.
Invasive ventilation	68 (59%)	62 (65%)	6 (32%)	0.007	61 (59%)	7 (58%)	n.s	60 (61%)	8 (47%)	n.s.
Renal replacement therapy	17 (15%)	16 (17%)	1 (5%)	n.s.	15 (15%)	2 (17%)	n.s	16 (16%)	1 (6%)	n.s.
AKI	68 (62%)	57 (62%)	11 (65%)	n.s.	61 (62%)	7 (64%)	n.s	60 (63%)	8 (62%)	n.s.
Severe AKI	19 (17%)	17 (18%)	2 (12%)	n.s.	18 (18%)	1 (9%)	n.s	19 (20%)	0 (0%)	n.s.
SARS-CoV-2 Plasma	57 (64%)	53 (64%)	4 (67%)	n.s.	52 (63%)	5 (83%)	n.s	50 (62%)	7 (88%)	n.s.
Days with symptoms on ICU arrival	10 (8–12)	10 (8–12)	10 (8–13)	n.s.	10 (9–12)	9 (7–11)	n.s.	10 (9–12)	9 (7–11)	n.s.
BMI	28.6 (25.6–33.2)	29.0 (26.6–33.4)	26.4 (22.9–29.2)	n.s.	28.6 (25.6–32.8)	28.7 (26.4–38.3)	n.s.	28.7 (26.2–33.4)	26.9 (22.9–32.3)	n.s.

Data are expressed as *n* (%) or median (interquartile range, IQR). Statistically significant differences between groups marked in red. Antibody-positive: Before ICU discharge. Abbreviations: Alive at 30 days: 30 days from ICU admission. AKI: Acute kidney injury. Severe AKI: AKI ≥ stage III. SARS-CoV-2 plasma: Patients with SARS-CoV-2 virus detected in plasma. BMI: Body mass index (kg/m^2^). n.s.: Not significant. Groups compared with Z-test or Mann-Whiney U test.

**Table 3 jcm-11-03419-t003:** Organ support in relation to positivity for anti-SARS-CoV-2 antibodies.

		IgM Positive		
	Total	Yes *n* = 103	No *n* = 12	*p*
ICU-free days	17 (0–24)	18 (0–24)	0 (0–0)	0.002
RRT-free days	30 (15–30)	30 (28–30)	0 (0–15)	<0.001
Ventilator-free days	24 (6–30)	25 (15–30)	0 (0–11)	0.002
Vasoactive-free days	25 (15–30)	26 (19–30)	0 (0–19)	0.002
Lowest *p*/f-ratio	78.8 (69.8–95.3)	78.8 (69.8–95.3)	76.5 (66.0–105.5)	n.s.
SARS-CoV-2 plasma (copies/mL)	0 (0–800)	0 (0–800)	600 (0–1100)	n.s.
		IgG positive		
	Total	Yes *n* = 98	No *n* = 17	*p*
ICU-free days	17 (0–24)	18 (0–23)	0 (0–26)	n.s.
RRT-free days	30 (15–30)	30 (28–30)	0 (0–30)	0.012
Ventilator-free days	24 (6–30)	25 (12–30)	11 (0–30)	n.s.
Vasoactive-free days	25 (15–30)	26 (19–30)	19 (0–30)	n.s.
Lowest *p*/f-ratio	78.8 (69.8–95.3)	78.0 (69.8–94.5)	89.3 (68.3–105.8)	n.s.
SARS-CoV-2 plasma (copies/mL)	0 (0–800)	0 (0–800)	450 (0–2000)	n.s.
		IgA positive		
	Total	Yes *n* = 96	No *n* = 19	*p*
ICU-free days	17 (0–24)	18 (0–23)	0 (0–27)	n.s.
RRT-free days	30 (15–30)	30 (26–30)	30 (0–30)	0.039
Ventilator-free days	24 (6–30)	24 (12–30)	21 (0–30)	n.s.
Vasoactive-free days	25 (15–30)	26 (19–30)	22 (0–30)	n.s.
Lowest *p*/f-ratio	78.8 (69.8–95.3)	78.0 (69.0)–93.0)	89.3 (70.5–113.3)	n.s.
SARS-CoV-2 plasma (copies/mL)	0 (0–800)	0 (0–800)	150 (0–2600)	n.s.

Data are expressed as median (interquartile range, IQR). Statistically significant differences between groups marked in red. Antibody-positive: Before ICU discharge. Abbreviations: RRT: Renal replacement therapy. Vasoactive-free days: Days without vasoactive treatment. *p*/f-ratio: mmHg/FiO_2_ SARS-CoV-2 plasma: Viral copies in plasma.

## Data Availability

Individual level data is available from the authors on reasonable request as detailed at https://doi.org/10.17044/scilifelab.14229410.v1.

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
