# Peer review of "Impaired Antibody Response Is Associated with Histone-Release, Organ Dysfunction and Mortality in Critically Ill COVID-19 Patients"

_jcm, 2022, doi:10.3390/jcm11123419_

Round 1
Reviewer 1 Report
In this study the authors report on the innate and adaptive immune responses in a single centre cohort of Covid positive ICU patients.
Minor concerns:
- References missing for similar studies (Janssen et al J. Infect Dis April 2021) (Nossent et al Front Immunol May 2021) (Cani et al Crit Care Explore Dec 2021). These reference will strengthen your discussion about how immune response are driven by NET formation could contribute to mortality and organ dysfunction.
- Although not significant it is interesting that patients with the higher Ig production also had evidence of thrombosis. Given NET production drives thrombosis and HistoneCit3 is associated with reduced endogenous anticoagulants what is the balance between the vigour of the innate immune response and subsequent adaptive immunity.
Author Response
Thank you for your valuable input. Please find below our response to your comments.
To facilitate an overview, the comments from the reviewers are in black font, our responses are in red font, the changes we have made in the manuscripts in green blue font and page numbers are in black bold font.
Minor concerns:
- References missing for similar studies (Janssen et al J. Infect Dis April 2021) (Nossent et al Front Immunol May 2021) (Cani et al Crit Care Explore Dec 2021). These reference will strengthen your discussion about how immune response are driven by NET formation could contribute to mortality and organ dysfunction.
We completely agree and the above references have now been included together with short clarifications in the discussion section.
Page 8, line 361, line 350-353 and line 378
- Although not significant it is interesting that patients with the higher Ig production also had evidence of thrombosis. Given NET production drives thrombosis and HistoneCit3 is associated with reduced endogenous anticoagulants what is the balance between the vigour of the innate immune response and subsequent adaptive immunity.
This is indeed an interesting pattern. However, it is hard to interpret non-significant results as you point out. Previous findings and ours would indicate that the immune response in severe COVID-19 increases the risk of thrombosis. If our findings of less thrombosis in antibody-negative patients despite their higher risk of a poor outcome would be confirmed, it would suggest that a poor adaptive immune response actually is protective against thrombosis in COVID-19. This could in that case be explained by a milder systemic immune response. Our study only includes patients in the ICU. Many previous studies have compared groups with both mild and sever COVID-19 and this fact might in some part explain some of the differences in the results. We have added two sentences in the discussion part about this finding but due to non-significance, we do not want to highlight it further, since it might lead to criticism once the article is published.
The higher rate of thrombosis in antibody-positive patients (non-significant) might be a reflection of a more powerful general systemic inflammatory response that clears virus more rapidly at the cost of a stronger pro-thrombotic state. However, this is highly speculative and not directly supported by our data.
Page 7, line 340-344

Reviewer 2 Report
This paper aims at finding a relation between the antibody response and the pathophysiology of severe cases of SARS-CoV-2. To this end, several data from the molecular to the clinical level were collected and compared with the antibody level. This study provides a wide set of parameters gathered for all the period of stay in intensive care. However, it is difficult to abstract a clear message from this study.
First of all this paper inspired of their previous work do not come to any new conclusions. It only demonstrates that a deficient antibody response leads to poor outcomes which has already been observed by their team and seems logical as they use it as their working hypothesis. Furthermore their objective that drives all the analyses lack of clarity, in fact neither a clear outcome to predict nor a relevant one is defined.
Moreover this approach seems pointless to me regarding Covid -19 clinical issue. They don’t find any marker for an early diagnostic of a poor outcome as the lack of antibody is measured at the end of the ICU stay and no clear sign of correlation with any parameters measured were found.
In terms of methodological choices, as the purpose of all experiments remains unclear, it seems they have been performed randomly in order to collect a maximum of data. As for the dosage of histones which came out of nowhere, no rational or scientific explanation justifies their interest or even why they are being sensed as remarkable in context of SARS-CoV-2. Another methodological apparent discrepancy is the detection of IgA in plasma. Indeed it is an antibody mainly found in secretion, then their dosage in plasma isn’t appropriate to determine if patients secrete them or not. It would have been more relevant to dose IgA in an other fluid.
Finally the choice of mortality at day 30 is debatable. Actually severe SARS-CoV-2 patients in the ICU die mostly after day 30, with a high proportion of treatment withdrawal in the context of multiple secondary infections. Therefore, mortality at day 30 is probably not relevant in this context. Authors should consider to analyse mortality at day 60 or day 90. In short they should consider to present this work in short communication or brief abstract.
Author Response
Thank you for your valuable input. Please find below the response to your comments.
To facilitate an overview, the comments from the reviewers are in black font, our responses are in red font, the changes we have made in the manuscripts in green blue font and page numbers are in black bold font.
This paper aims at finding a relation between the antibody response and the pathophysiology of severe cases of SARS-CoV-2. To this end, several data from the molecular to the clinical level were collected and compared with the antibody level. This study provides a wide set of parameters gathered for all the period of stay in intensive care. However, it is difficult to abstract a clear message from this study.
We thank the reviewer for this comment. Indeed, our study includes a number of analyses of both survival, organ dysfunction and immune response. We consider that our aims in the introduction (to investigate antibody responses in critically ill patients with COVID-19 in relation to inflammation, organ failure and 30-day survival) is addressed in the abstract (In patients with severe COVID-19 requiring intensive care, a poor antibody response is associated with organ failure, systemic histone release and increased 30-day mortality.) as well as in the results section. However, after your input, we have clarified some of the sentences to specify our aim.
Introduction: The aim of this study is to describe the effects of an impaired antibody response, with regards to 30-day survival, organ failure and activation of other parts of the immune system, identifying key pathophysiological mechanisms in a large group of intensive care patients.
Page 2, line 80-83
Conclusion, abstract: In patients with severe COVID-19 requiring intensive care, a poor antibody response is associated with organ failure, systemic histone release and increased 30-day mortality.
Page 1, line 41-43
Conclusion, main: This study confirms that a weak anti-SARS-CoV-2 antibody response in intensive care patients with COVID-19 is associated with increased 30-day mortality and our re-sults suggest that this may be due to multiple organ dysfunction and NETosis.
Page 9, line 419-421
First of all this paper inspired of their previous work do not come to any new conclusions. It only demonstrates that a deficient antibody response leads to poor outcomes which has already been observed by their team and seems logical as they use it as their working hypothesis. Furthermore their objective that drives all the analyses lack of clarity, in fact neither a clear outcome to predict nor a relevant one is defined.
We understand the point the reviewer is making. As you correctly point out, we previously published an article on this subject (Reference 26). However, the first article was published as a letter and only included 19 patients. We consider it reasonable to try to confirm interesting findings in small studies in larger studies, which we now have done in this manuscript. We have made an effort to describe our aims in detail and have done our best to address them in the conclusion. Furthermore, in this study we also included considerably more patients and analysed several aspects of the immune system, in order to describe underlying mechanisms for our findings. Indeed, in some parts our results are only hypothesis generating, which we try to point out in our discussion part. Furthermore, we never aimed at creating a prediction model, this study is more related to identifying potential mechanisms of disease. In general, we have tried not to over interpret our results and discussed the limitations of the study.
Moreover this approach seems pointless to me regarding Covid -19 clinical issue. They don’t find any marker for an early diagnostic of a poor outcome as the lack of antibody is measured at the end of the ICU stay and no clear sign of correlation with any parameters measured were found.
As you correctly point out, antibodies were measured as late as possible to maximise the chance for them to form and be measurable, as described in the methods section. If we would have analysed antibody levels at admission it is very likely that we would have had many false negative results, since it takes time for the antibodies to form. Although it is of great interest, our primary aim with this investigation was not to find an early diagnostic of a poor outcome. The fact that we did not find correlations to several of the components of the immune system is indeed correct, but nothing we could have anticipated when designing the study. However, we found a correlation between poor antibody response and survival and organ dysfunction in a large group of severely ill COVID-19 patients, a finding that we hope you find interesting.
In terms of methodological choices, as the purpose of all experiments remains unclear, it seems they have been performed randomly in order to collect a maximum of data. As for the dosage of histones which came out of nowhere, no rational or scientific explanation justifies their interest or even why they are being sensed as remarkable in context of SARS-CoV-2. Another methodological apparent discrepancy is the detection of IgA in plasma. Indeed it is an antibody mainly found in secretion, then their dosage in plasma isn’t appropriate to determine if patients secrete them or not. It would have been more relevant to dose IgA in an other fluid.
We appreciate this comment from the reviewer. We have now added a section in the introduction as a complement to the discussion section, highlighting why we are discussion NET-formation to make it clearer.
Several groups have also reported that SARS-CoV-2 can induce neutrophil extracellular trap (NET) formation in neutrophiles and that NET-formation could be part of the immunopathology in severe COVID-19.[19-25]
Page 2, line 74-77
We completely agree that analyses of IgA in secretions would have been an interesting complementary analysis, but unfortunately, we did not have any other fluid besides plasma for analysis. However, the findings of IgA in plasma at least proves that the immune system produces IgA, a finding that is not unimportant, and thus we consider it appropriate to report, together with the other antibody subclasses.
Finally the choice of mortality at day 30 is debatable. Actually severe SARS-CoV-2 patients in the ICU die mostly after day 30, with a high proportion of treatment withdrawal in the context of multiple secondary infections. Therefore, mortality at day 30 is probably not relevant in this context. Authors should consider to analyse mortality at day 60 or day 90. In short they should consider to present this work in short communication or brief abstract.
We thank the reviewer for this excellent suggestion. We have analysed mortality after 90 days in relation to antibody levels and added it to results section. This is added in Table 2.
Patients positive for anti-SARS-CoV2 antibodies had higher 30-day survival (which was the main outcome in the present analysis) compared to patients negative for antibodies (30-day survival for IgM 83% vs 33%, IgG 82% vs 53% and IgA 83% vs 47%). As a complementary analyses, 90-day survival was analysed. This confirmed the findings with higher survival rates in the patient groups positive for anti-SARS-CoV2 antibodies.
Page 5, line 197-199. Please notice that Table 2 on page 7 is also updated with 90-day survival.

Reviewer 3 Report
The study by Lagedal and colleagues aims to assess the effect of antibody response in critically ill COVID-19 patients. The main finding of the study is that higher levels of anti-SARS-CoV 2 antibodies, especially IgM, are associated with better outcome in critically ill COVID-19 patients.
The authors should address the following points:
Did patients receive any kind of pharmacological treatment for COVID-19 (e.g. corticosteroids)? If yes, this should be added and, possibly, taken into consideration in the interpretation of the results.
Lines 201-204: the correlations reported are very weak. This should be clearly stated in the text. If the authors believe that this is major finding of the study, a figure showing the graphical representation of these correlations should be added in the main document.
Antibody-positive COVID-19 patients, who were shown to have better outcomes, had higher CRP levels, a trend towards lower d-dimer levels, no significant differences in all cytokine levels and higher complement factors levels. All of these inflammatory and thrombotic markers have repeatedly been shown to correlate with worse outcomes in COVID-19 and have even been used to guide anti-inflammatory therapy or as therapeutic targets per se in this patient population. How do authors explain this contradiction?
Lines 322-325: Please provide references showing thrombocytosis in severe COVID-19 patients.
Minor:
Materials and methods section: provide details about the ELISA methodology used for C3a and C5b-9 measurements.
Line 26: spell out ICU.
Table 3. please report PFR values according to the Berlin definition of ARDS, e.g. a patient with PaO2 80 while breathing FiO2 80%, will have a PFR of 80/0.8 = 100 mmHg
Author Response
Thank you for your valuable input. Please find below our response to your comments.
To facilitate an overview, the comments from the reviewers are in black font, our responses are in red font, the changes we have made in the manuscripts in green blue font and page numbers are in black bold font.
The study by Lagedal and colleagues aims to assess the effect of antibody response in critically ill COVID-19 patients. The main finding of the study is that higher levels of anti-SARS-CoV 2 antibodies, especially IgM, are associated with better outcome in critically ill COVID-19 patients.
The authors should address the following points:
Did patients receive any kind of pharmacological treatment for COVID-19 (e.g. corticosteroids)? If yes, this should be added and, possibly, taken into consideration in the interpretation of the results.
We thank the reviewer for this interesting question. As you point out this would have been important to report. However, the patients in this study originate from the first wave of COVID-19 in Sweden and no specific treatments were used at this time (i.e. IL-6 antagonists or corticosteroids).
Lines 201-204: the correlations reported are very weak. This should be clearly stated in the text. If the authors believe that this is major finding of the study, a figure showing the graphical representation of these correlations should be added in the main document.
Thank you for noticing! As you point out, the correlations are weak and this is now highlighted in the discussion section.
Furthermore, patients with poor antibody response have higher rates of organ failure based on SAPS3 and SOFA score, even if the correlations are weak and should be interpreted with caution, and finally on the duration of organ support such as renal replacement therapy and vasoactive medication.
Page 6, line 281-285
Antibody-positive COVID-19 patients, who were shown to have better outcomes, had higher CRP levels, a trend towards lower d-dimer levels, no significant differences in all cytokine levels and higher complement factors levels. All of these inflammatory and thrombotic markers have repeatedly been shown to correlate with worse outcomes in COVID-19 and have even been used to guide anti-inflammatory therapy or as therapeutic targets per se in this patient population. How do authors explain this contradiction?
We agree with the reviewer that this is an interesting observation. When interpreting the results we have discussed the following:
In the very small sub-group of patients with no antibody response there may be a generally impaired immune response that contributes to their increased mortality. This suggests that the lack of antibodies is merely a signal of a defect immune system, not a cause of increased mortality per se. If so it would be logical that these patients have lower levels of CRP, d-dimer and complement factors. However, in a larger cohort of patients (often including both general ward and ICU patients) the group of immunological impaired patients highlighted in this study (that lacks antibody response) is not large enough to make an impact on the outcome of the entire cohort, where an excessive inflammatory response is associated with worse outcomes. This is merely speculation and not something we have investigated in this study, so we have so far chosen not to include it in the discussion.
Lines 322-325: Please provide references showing thrombocytosis in severe COVID-19 patients.
As you point out this is not totally correct. This section is now corrected, and a new reference is added. It seems that most COVID-19 patients have normal platelet counts and that low numbers are associated with worse outcome.
In contrast to other systemic infections such as bacterial sepsis that are associated with low platelet counts due to consumption, many patients with severe COVID-19 typically display normal platelet counts. An association between low platelet counts and higher mortality in COVID-19 is previously described.[44]
Page 7, line 329-333
Minor:
Materials and methods section: provide details about the ELISA methodology used for C3a and C5b-9 measurements.
A section has been added in the m&m section.
C3a by ELISA using monoclonal antibody (mAb) 4SD17.3 for capture and biotinylated polyclonal anti-C3a for detection and sC5b-9 by an in-house ELISA using anti-human neo-C9 mAb aE11 for capture and polyclonal anti-C5 for detection.
Page 4, line 160-162
Line 26: spell out ICU.
Corrected
Table 3. please report PFR values according to the Berlin definition of ARDS, e.g. a patient with PaO2 80 while breathing FiO2 80%, will have a PFR of 80/0.8 = 100 mmHg
This has now been adjusted to mmHg. Please observe that the changes now made in the table are not marked by red since it would interfere with the significant findings.
Table 3

Round 2
Reviewer 2 Report
I thank the authors for their comment and responses